# Recombinant *Escherichia coli* BL21 with LngA Variants from ETEC E9034A Promotes Adherence to HT-29 Cells

**DOI:** 10.3390/pathogens12020337

**Published:** 2023-02-16

**Authors:** Karina Espinosa-Mazariego, Zeus Saldaña-Ahuactzi, Sara A. Ochoa, Bertha González-Pedrajo, Miguel A. Cevallos, Ricardo Rodríguez-Martínez, Mariana Romo-Castillo, Rigoberto Hernández-Castro, Ariadnna Cruz-Córdova, Juan Xicohtencatl-Cortes

**Affiliations:** 1Unidad de Enfermedades Infecciosas, Laboratorio de Investigación en Bacteriología Intestinal, Hospital Infantil de México Federico Gómez, Ciudad de México 06720, Mexico; 2Departamento de Genética Molecular, Instituto de Fisiología Celular, Universidad Nacional Autónoma de México, Ciudad de México 04510, Mexico; 3Centro de Ciencias Genómicas, Programa de Genómica Evolutiva, Universidad Nacional Autónoma de México, Cuernavaca 62210, Mexico; 4CONACYT-Laboratorio de Investigación en Bacteriología Intestinal, Hospital Infantil de México Federico Gómez, Ciudad de México 06720, Mexico; 5Departamento de Ecología de Agentes Patógenos, Hospital General Dr. Manuel Gea González, Ciudad de México 14080, Mexico

**Keywords:** ETEC, CS21 pilus, HT-29 cells, adherence assays, point mutation

## Abstract

The CS21 pilus produced by enterotoxigenic *Escherichia coli* (ETEC) is involved in adherence to HT-29 intestinal cells. The CS21 pilus assembles proteins encoded by 14 genes clustered into the *lng* operon. Aim. This study aimed to determine whether *E*. *coli* BL21 (ECBL) transformed with the *lng* operon lacking the *lngA* gene (pE9034AΔ*lngA*) and complemented *in trans* with *lngA* variants of ETEC clinical strains, as well as point substitutions, exhibited modified adherence to HT-29 cells. Methods. A kanamycin cassette was used to replace the *lngA* gene in the *lng* operon of the E9034A strain, and the construct was transformed into the ECBL strain. The pJET1.2 vector carrying *lngA* genes with allelic variants was transformed into ECBLpE9034AΔ*lngA* (ECBLΔ*lngA*). The point substitutions were performed in the pJET*lngA*_FMU073332_ vector. Results. Bioinformatic alignment analysis of the LngA proteins showed hypervariable regions and clustered the clinical ETEC strains into three groups. Variations in amino acid residues affect the adherence percentages of recombinant ECBL strains with *lngA* variants and site-specific mutations with HT-29 cells. Conclusion. In this study, ECBL carrying the *lng* operon harboring *lngA* variants of six clinical ETEC strains, as well as point substitutions, exerted an effect on the adherence of ECBL to HT-29 cells, thereby confirming the importance of the CS21 pilus in adherence.

## 1. Introduction

Enterotoxigenic *Escherichia coli* (ETEC) is responsible for nearly 220 million cases of diarrhea annually, and approximately 75 million occur in children under five years of age with 18,700 deaths according to the “Institute for Health Metrics and Evaluation” estimates and 42,000 deaths according to “Maternal-Infant Epidemiology” estimates [1,2]. ETEC-induced intestinal infections in children less than 5 years of age are particularly common among low-income populations and in areas with poor sanitary conditions [3,4]. ETEC is responsible for many cases of traveler’s diarrhea worldwide, and the clinical symptoms include watery diarrhea induced by secretion of a thermolabile toxin (LT) and/or a thermostable toxin (ST) [5,6]. STa and LT-I are associated with the disease in both humans and animals, STb is mainly associated with diarrhea in piglets, and LT-II has only been associated with animal disease [7].

The colonization by ETEC of its host is mediated by fimbrial and nonfimbrial adhesins, which show antigenic differences. Colonization Factors (CFs) contribute to the capacity of bacteria to efficiently colonize gut enterocytes via specific cell receptors [8,9]. In vivo studies have demonstrated that the ETEC FMU073332 strain efficiently colonizes the mouse intestine and exhibits tropism for the mouse ileum [10]. The CS21 pilus is one of the most prevalent CFs in clinical ETEC strains worldwide [9]. ETEC strains that produce the CS21 pilus have been associated with adhesion to intestinal epithelial cells, self-aggregation, and twitching motility [11,12,13]. Other studies using a neonatal murine model have shown that the CS21 pilus contributes to the pathogenesis of ETEC in vivo [14].

The CS21 pilus of ETEC is mainly composed of hundreds of repeats of the LngA protein, and the interactions among these proteins favor the assembly of homopolymeric structures with a diameter of 7 nm and a length greater than 20 μm [11,13,15]. The CS21 pilus is located at the poles of the bacterial cell surface, and its assembly is mediated by the expression of the LngA protein, which is influenced by nutritional factors, bacterial growth, osmolarity, and pH [11]. In response to different environmental signals, the expression of CS21 involves both local and global regulators, such as H-NS and CRP [16]. The *lngA* gene encodes the LngA protein, is contained in a plasmid of 14 kb and shares 74–79% sequence identity with the CofA protein of the CFA/III pilus of ETEC [13,15,17,18]. Crystallographic data of the C-terminus of the CofA protein show a highly variable loop with exposed amino acid residues, which could be involved in recognition of the protein by its receptor in host cells [18]. A three-dimensional (3D) model of the LngA protein generated from the CofA crystal structure shows the presence of variable amino acid residues in the C-terminal region, and these residues are likely involved in interactions with specific receptors of intestinal epithelial cells [18].

We recently demonstrated that the LngA, LngB, LngC, LngD, LngH, and LngP proteins are essential for CS21 assembly as well as for bacterial aggregation and adherence to HT-29 intestinal cells [13]. The predicted subcellular localization of CS21 proteins is similar to that of well-known proteins of other type IV pilus homologs. Additionally, these data suggest that the LngB protein is localized at the tip of CS21. The LngA protein is processed by the LngP prepilin peptidase and another less efficient peptidase that has yet to be described [13]. The *lngA* gene is commonly found to be highly expressed in clinical ETEC strains from different parts of the world, such as Egypt, Bangladesh, Argentina, Chile, and Mexico [3,4,12,19]. These clinical strains, some that were recently included in a collection of *lngA* gene-positive clinical ETEC strains from Mexico and Bangladesh, have shown different adherence profiles [12]. The genomic sequence of the ETEC strain FMU073332, which belongs to sequence type 4 and clonal complex 10, has been described, and genes encoding CFs and toxins (*eltA*, *eltB*, *sta*2, *cstH*, *lngA, etpA, etpB,* and *etpC*) have been identified [20]. A phylogenetic tree revealed that three distinct branches (1, 2, and 3) are associated with the variability of the *lngA* gene in clinical ETEC strains [17]. A recent study showed that the different adherence profiles identified in clinical ETEC strains from Mexico and Bangladesh could be due to the high genetic variability of *lngA* [12]. The aim of this study was to determine the adherence effect of *E. coli* BL21 (ECBL) on HT-29 cells when transformed with six *lngA* variants from clinical ETEC strains.

## 2. Materials and Methods

### 2.1. Bacterial Strains and Growth Conditions

A collection of 12 ETEC clinical strains previously characterized as weakly, moderately, and strongly adherent were included in this study for *lngA* gene sequence analysis (Table 1). *E. coli* BL21 (ECBL) and E9034AΔ*lngA*::km were obtained from the laboratory collection as genetic templates for transformation assays (Table 2). The role of the CS21 pilus was evaluated in the ECBL strain by transformation with the *lngA* gene from different ETEC clinical strains. The sequences of *lngA* from E9034A and ETEC FMU73332 were employed for site-directed mutagenesis assays. The bacterial strains were stored at −70 °C in Luria Bertani (LB, Difco, Franklin Lakes, NJ, USA) broth supplemented with 20% glycerol (*v*/*v*). The clinical ETEC or recombinant ECBL strains were cultured in pleuropneumonia-like organism (PPLO) broth (BD Difco, Franklin Lakes, NJ, USA) to induce expression of the CS21 pilus [12,16,20]. LB or PPLO broth medium was previously supplemented with kanamycin (50 μg/mL) and/or ampicillin (100 μg/mL) as needed.

### 2.2. Sequencing of the lngA Gene from Different Clinical ETEC Strains

The *lngA* gene with its promoter site from the 12 clinical ETEC strains was amplified by endpoint polymerase chain reaction (PCR) using the enzyme Platinum^®^ Taq DNA Polymerase High Fidelity^®^ (Thermo Fisher Scientific, Carlsbad, CA, USA). The specific amplicons were subjected to electrophoretic analysis with 1.5% agarose gels and stained with 0.05% ethidium bromide solution. Subsequently, the DNA products were purified and sequenced by LANGEBIO (Institute of Genomic Services “LANGEBIO-CINVESTAV”, Irapuato Guanajuato, Mexico) using the Illumina platform and analyzed using the Chromas bioinformatic program (http://technelysium.com.au/wp/chromas/ accessed on 1 September 2022 ). Sequence translation was used to determine the LngA protein sequence (https://web.expasy.org/translate/ accessed on 5 September 2022).

### 2.3. Cloning of the lngA Variants into the pJET1.2 Vector

DNA of the clinical ETEC strains was obtained from 200 µL of a bacterial suspension grown in LB broth to an OD_600_ of 1.0. Briefly, the bacteria were resuspended in distilled water, boiled for 5 min, and centrifuged for 5 min at 25 g to obtain the total DNA. The *lngA* variants were amplified by PCR using Platinum^R^ *Taq* DNA Polymerase High Fidelity (Invitrogen, Carlsbad, CA, USA) and forward- and reverse-specific primers (Table 3). The *lngA* variants were cloned into the pJET1.2 vector following the instructions provided by the manufacturer (Thermo Fisher Scientific; Carlsbad, CA, USA) (Table 4). The DNA variant product cloned into the pJET1.2 vector was sequenced using forward and reverse primers at the Institute of Genomic Services “LANGEBIO-CINVESTAV” Campus Irapuato, Mexico.

### 2.4. Transformation of ECBL with the lng Operon and lngA Variants

The 14-kb plasmid harboring the *lng* operon of the CS21 pilus was purified from the ETEC E9034A∆*lngA* strain [20] using a ZymoPURE™ II Plasmid Maxiprep Kit (Zymo Research, Irvine, CA, USA). The *lngA* gene was previously replaced with a kanamycin cassette using the one-step inactivation method [23]. Mobilization of the plasmid into ECBL was performed via electroporation (BMC Harvard Apparatus, Cambridge, MA, USA) at 1800 V, and the transformed strains were selected by overnight culture on LB agar plates with kanamycin. The positive colonies were named ECBLpE9034AΔ*lngA* (ECBLΔ*lngA*). The DNA product of the ECBLΔ*lngA* strain was confirmed by PCR using specific primers for the *lngA*, *lngD*, and *lngH* genes (Table 3) and by sequencing. In addition, electrocompetent cells derived from the ECBLΔ*lngA* strain were transformed by electroporation with 100 ng of the pJET1.2 plasmid, which carried variants of the *lngA* gene of the clinical ETEC strains selected for this study (Table 4). The transformed colonies were considered positive when cultured on LB agar plates with both kanamycin and ampicillin.

### 2.5. Site-Directed Mutagenesis of the lngA Gene

In this study, the LngA sequence obtained from the ETEC FMU073332 strain was selected for site-directed mutagenesis. In a previous study, ETEC E9034A, FMU073332, and strains with isogenic mutations in the *lngA* gene were compared according to their adhesion ability. ETEC FMU073332 was more adherent than E9034A [12]; and complete genome data are available for ETEC FMU073332 (NZ_CP017844.1). To identify the variability in the amino acid residues, the FMU07332 LngA variant was cloned into the pJET 2.0 vector to generate pJET*lngA*_FMU073332_, and following the QuikChange protocol provided by Stratagene, the vector was transformed into ECBLΔ*lngA* to restore the *lngA* gene with specific mutations.

Point substitutions of the amino acid residues in the sequence of the _LngAFMU073332_ protein based on the LngA_E9034A_ protein were performed according to terminal carboxyl variability, as presented in detail in Table 5. All constructs were verified by DNA sequencing.

### 2.6. Genome Sequencing

The genomic DNA of the ECBLΔ*lngA*/p_FMU073332_ strain was extracted using a genomic DNA purification kit (Thermo Fisher, NY, USA). The DNA was used to construct an Oxford Nanopore library following the Genomic DNA by Ligation protocol (SQK-LSK109) (Oxford Nanopore Technologies, ONT). The DNA was not sheared but rather used directly for subsequent steps ranging from purification to library construction. Reads were obtained with the MinION ONT device using the MinION R9.4.1 flow cell. In total, 164,952 reads were obtained. Base calling was performed using Guppy software (v4.0.14), and adapter sequence removal was performed with Porechop (v0.2.4; https://github.com/rrwick/Porechop accessed on 8 February 2023). The raw reads were aligned against an *in silico* reconstruction of the ECBLΔ*lngA*/pJET*lngA*_FMU073332_ (ECBLΔ*lngA*/p_FMU073332_) genome with bowtie2 (2.3.4.1) [24]. The read mappings were visualized using the Artemis genome visualization tool [25].

### 2.7. Reverse Transcription Polymerase Chain Reaction (RT–PCR)

Total RNA was extracted after all strains (wild type, recombinant ECBL strains with *lngA* variants, and recombinant ECBL strains with site-specific *lngA* mutations) were grown overnight in LB medium using an RNeasy Mini Kit (QIAGEN, Hilden, NRW, Germany) and treated with DNase I according to the manufacturer’s instructions (Invitrogen, Carlsbad, CA, USA). RT–PCR was performed using the OneStep RT–PCR Kit (QIAGEN, Hilden, NRW, Germany). The RT–PCR conditions were as follows: one cycle of 60 °C for 30 min, one cycle of 95 °C for 15 min, 30 cycles of 95 °C for 1 min, 62 °C for 1 min, and 72 °C for 1 min, and one final extension at 72 °C for 10 min. The RT–PCR products were electrophoresed in 1.5% agarose gels (Promega, Madison, WI, USA) using 1% TAE buffer (Tris-acetate-EDTA). The specific primers used in the RT–PCR assays are described in Table 3. The 16S gene was used as an internal control.

### 2.8. SDS–PAGE and Western Blot Analyses

The bacterial strains were resuspended in 1× Laemmli buffer, boiled for 5 min, and subjected to 16% sodium dodecyl sulfate–polyacrylamide gel electrophoresis (SDS–PAGE) [26]. The separated proteins contained in the acrylamide gels were transferred onto polyvinylidene fluoride (PVDF) membranes. The membranes were blocked for 1 h with 5% skim milk in phosphate-buffered saline (PBS), pH 7.4, with 0.5% Tween-20, and the blocked membranes were incubated for 1 h with anti-CS21 polyclonal rabbit or anti-DnaK monoclonal antibodies (MBL International, Woburn, MA, USA) as a loading control. The membranes were washed 3 times with PBS-Tween 0.5% and incubated for 1 h with goat anti-rabbit IgG conjugated to horseradish peroxidase (Sigma-Aldrich Co., St. Luis, MO, USA). The membranes were washed 5 times with 0.5% PBS-Tween and revealed by chemiluminescence-ECL (Amersham Life Science; Arlington Heights, IL, USA). All strains were included in this assay (wild type, recombinant ECBL strains with *lngA* variants, and recombinant ECBL strains with site-specific *lngA* mutations).

### 2.9. Assay of Adherence to HT-29 Cells

HT-29 intestinal cells (ATCC HTB-38) were seeded into 24-well plates (Corning^®^ Costar^®^, Corning, NY, USA) in 1 mL of Dulbecco’s modified Eagle medium (DMEM; Gibco, Invitrogen USA) until a cellular monolayer with 80% confluence (~1 × 10^5^ cells) formed. The cell monolayers were infected with 1 × 10^7^ colony forming units (CFUs)/mL ETEC (E9034A, 48342, 62123, 63880, 64760, 44166, 45162, 49247, FMU073332, 63280, 63283, and 45163) strains, recombinant ECBL strains with *lngA* variants, and recombinant ECBL strains with site-specific *lngA* mutations (bacteria were previously cultured in PPLO broth overnight at 37 °C) and incubated for 6 h at 37 °C in an atmosphere with 5% CO_2_. The culture medium was removed after infection, and the attached bacterial cells were washed three times with 1 mL of 1× PBS. The bacteria that adhered to the cell monolayers were detached with 1 mL of 0.1% Triton (Amresco Bioscience, Solon, OH, USA) in 1× PBS and serially diluted (10^−1^ to 10^−5^), and 10 microliters of each sample was spread on LB agar plates supplemented with antibiotics as needed; thereafter, the plates were incubated at 37 °C overnight. The bacterial adherence was analyzed quantitatively by determining the CFUs per mL and qualitatively by light microscopy (staining with 1% Giemsa).

### 2.10. Statistical Analysis

The data generated from the quantitative analysis of the adherence assays were analyzed using unpaired Student’s *t*-test. In all analyses, a *p* value less than or equal to 0.05 was considered to indicate statistical significance.

## 3. Results

### 3.1. Sequencing of the lngA Gene from Different Clinical ETEC Strains

The *lngA* gene with its promoter site was amplified from a collection of 12 clinical ETEC strains. The PCR products were purified and sequenced, the nucleotide sequences were translated (https://www.ebi.ac.uk/Tools/st/ accessed on 1 September 2022), and the 12 amino acid sequences without promoters were aligned (http://multalin.toulouse.inra.fr/multalin/ accessed on 5 September 2022) (Figure 1). The *lngA* promoter regions were also analyzed, and the RNA polymerase-binding sites were determined to be at positions −10 and −35. The alignment showed five nucleotide variations, and no other changes were detected in the RNA–polymerase binding site (Appendix A). The nucleotide and amino acid sequence data are available at https://github.com/ariadnnacruz/CS21-LngA.git (accessed on 5 December 2022).

### 3.2. Identification of Variable Regions in the LngA Sequences from Clinical ETEC Strains

The amino acid sequences of the LngA protein from clinical ETEC strains (E9034A, 48342, 62123, 63880, 64760, 44166, 45162, 49247, FMU073332, 63280, 63283, and 45163) showed 17 variable regions at positions 15, 78, 88, 94, 97, 100, 108 to 109, 147, 178, 180, 182, 191 to 193, 202 to 204, 208, 211 to 212, 215, and 220 to 238 (Figure 1). The amino acid residues at positions 178 to 238 located in the C-terminus contain the main variable regions (Figure 1). In accordance with the amino acid variability in the LngA proteins, three groups were generated: clinical ETEC strains belonging to group V1 (E9034A, 48342, 62123, 63880, and 64760), group V2 (FMU073332, 45162, 49247, 63280, 63283, and 45163), and group V3 (44166) (Figure 1).

According to these data, six clinical ETEC strains (E9034A, FMU073332, 44166, 45162, 48342, 63280) showing variability in their amino acid sequences and different adherence profiles (weakly, moderately, and strongly adherent) were selected for further assays (Table 1).

### 3.3. Transcription of Recombinant ECBL Strains with lngA Variants, Site-Specific lngA Mutations and Expression of the lngA Gene

The ECBL strain carrying a plasmid that harbored the *lng* operon with the replacement of the *lngA* gene with a kanamycin cassette (ECBLpE9034AΔ*lngA*) was complemented with the pJET1.2 vector carrying the *lngA* variant plus 352 bp upstream of its ATG start codon, including the promoter region of the clinical ETEC strains. Six recombinant ECBL strains with *lngA* variants were obtained: ECBLΔ*lngA*p_E9034A_, ECBLΔ*lngA*p_FMU073332_, ECBLΔ*lngA*p_44166_, ECBLΔ*lngA*p_45162_, ECBLΔ*lngA*p_48342_, and ECBLΔ*lngA*p_63280_ (Table 4). Amplification of the *lngA* transcript yielded a 609-bp product, and a 22-kDa protein, which corresponded to the LngA protein, was confirmed by Western blot (WB) assays after incubation with anti-CS21 (Appendix A).

Site-specific mutations in the *lngA* sequence of FMU073332 were generated according to the C-terminal variation after alignment analysis (some amino acids were replaced by those found in ETEC E9034A) to determine whether these changes affected the ability to adhere to intestinal cells (Figure 2). ECBLpE9034AΔ*lngA* was transformed with pJET*lngA*_FMU073332_ vector variants to generate eight recombinant ECBL strains with site-specific *lngA* mutations: ECBLΔ*lngA*p_QTA-TAT_, ECBLΔ*lngA*p_GN-TT_, ECBLΔ*lngA*p_T-S_, ECBLΔ*lngA*p_G-S_, ECBLΔ*lngA*p_TD-NN_, ECBLΔ*lngA*p_K-T_, ECBLΔ*lngA*p_ER-DK_, and ECBLΔ*lngA*p_T-Q_ (Table 5). For recombinant ECBL strains with *lngA* variants, transcription and expression were confirmed by RT–PCR and WB assays (Appendix A).

### 3.4. Adherence to HT-29 Intestinal Cells Was Restored in Recombinant ECBL Strains with lngA Variants

Clinical ETEC strains and recombinant ECBL strains with *lngA* variants did not show differences in adherence values, independent of the group assignment based on amino acid sequences. ECBLΔ*lngA* was less adherent than E9034A and FMU073332 but showed stronger adherence than the ECBL strain. Almost all of the recombinant ECBL strains with *lngA* variants showed restored adherence, except two strains that exhibited an increase in adherence compared with the ETEC strains (ECBLΔ*lngA*p_48342,_ ECBLΔ*lngA*p_63280_ vs. ETEC 48342 and ETEC 63280) (Figure 3). In addition, mannose inhibition assays were performed to demonstrate that adherence was mediated mainly by LngA and not by type 1 fimbriae. The results showed a decrease in adherence with 1% mannose (*p* = 0.00048; Appendix A), compared with the results obtained from assays without mannose. Our data indicate the existence of other potential adhesins in the ECBL strain, such as the type 1 fimbriae involved in adherence to HT-29 intestinal cells. The qualitative analysis by light microscopy showed a correlation with data generated by the quantitative analysis (Figure 4).

### 3.5. Recombinant ECBL Strains with Site-Specific Mutations Showed Modified Adherence to HT-29 Intestinal Cells

CS21 pili are oligomers composed of thousands of copies of the LngA protein; we hypothesized that specific amino acids play an important role in the levels of adherence of ETEC strains to HT-29 intestinal cells. Specific substitution of the amino acid residues in the C-terminus of the LngA protein of the FMU073332 strain to amino acid residues in the LngA protein from the E9034A strain was performed to determine their effects on adherence to HT-29 intestinal cells. A decrease in adherence was observed only in four strains with site-specific mutations. This decrease was more evident in the ECBLΔ*lngA*p_T-Q_ strain harboring the substitution of T by Q, which decreased adherence to 86.35%. Analysis of the other strains revealed that reductions were observed with the following substitutions: GN by TT (36.42%), G by S (59.52%), and TD by NN (37.5%) (Figure 5). The qualitative analysis confirmed that the adherence among the clinical ETEC and recombinant ECBL strains with site-specific *lngA* mutations was the same (Figure 6).

ECBLΔ*lngA*p_FMU073332_ was sequenced by Nanopore technology, and a 14-kb plasmid was found to contain the *lng* operon. The *lngA* gene was deleted and replaced in this operon by a kanamycin resistance cassette. Moreover, the sequence of pJET1.2 harboring the *lngA* gene from FMU073332 was identified (PRJNA774369, link: https://www.ncbi.nlm.nih.gov/sra/PRJNA774369 accessed on 8 February 2023), and these data are consistent with the PCR results (Appendix A).

## 4. Discussion

The CS21 pilus from ETEC has been described as an adhesin that mediates the interaction among ETEC strains and intestinal cells and induces the formation of bacterial aggregates that protect them from antimicrobial compounds [11,12,27]. A collection of CS21-producing clinical ETEC strains from Mexican and Bangladeshi children with diarrhea were described as low, moderate, and strong adherents. The ETEC FMU073332 strain exhibits stronger adherence to HT-29 cells and shares its classic virulence factors [toxins (LT and ST) and CFs (CS3 and CS21)] with E9034A, as described by genome analysis [12,20].

In this study, the different adherence levels of clinical ETEC strains to HT-29 intestinal cells, as previously characterized, could be attributed to a repertoire of several CFs, including the CS21 pilus [12]. The *lngA* gene from 12 strains was selected and sequenced for this assessment. The *lngA* nucleotide sequence was translated to an amino acid sequence (234 to 236). Other studies have revealed that the sequences of the *lngA* gene of clinical ETEC strains have a length of 621 nucleotides, translated to 206 amino acid residues [15,17]. The difference in length between the LngA protein sequences is due to the inclusion of a signal peptide of 30 amino acid residues at the N-terminal end, and the prepellin peptidase removes this peptide to generate a mature protein [28]. Because each pilus filament is composed of thousands of pilin subunits, small differences in surface-exposed residues can significantly impact the affinity to its receptor on intestinal cells [29]. An analysis of different sequences of the PilA protein, which is the structural subunit of the type IV pilus of *Neisseria meningitidis* and *Neisseria gonorrhoeae*, has shown that this protein is divided into three regions: a conserved region (N-terminal), a semivariable region (middle part of the sequence), and a hypervariable region (C-terminal) [30]. We also found similar results for the amino acid sequences of the LngA proteins, probably because two regions were well defined (78 to 108 and 109 and 114), and the C-terminal region was the main variable region. According to 3D structure models of CofA of the CS8 pilus, the corresponding region from 189 to 236 of LngA has an exposed surface, which can participate specifically in interactions with some cellular ligands that have not been described [18,20].

This variability maintains a correlation with the pilin of the other type IV pili [28]. Gomez-Duarte et al. [17] classified the *lngA* genes into three distinct phylogenetic groups with 103 to 145 mutational events. Group V1 and V2 allelic variants were identified in this study, but another research group demonstrated that the antigenic diversity of *lngA* indicates significant structural conservation between the group variants [17]. According to these data, we selected six representative strains from the three groups of trees that exhibited different levels of adherence to clone the *lngA* sequence into ECBLΔ*lngA*-generated recombinant strains with *lngA* variants. The challenge was to demonstrate whether the CS21 pilus could be assembled in strains different from ETEC and whether their ability to adhere was maintained or modified.

The presence of specific amino acid variations in the LngA protein could modify the stability or affinity of the CS21 pilus based on different levels of ETEC adherence to HT-29 cells. However, only two recombinant ECBL strains with *lngA* variants displayed an increase in adherence compared with the ETEC strain, and both of these strains belonged to the same subgroup. Clinical ETEC strains from patients with diarrhea in Mexico and Bangladesh have shown different levels of adherence, and these differences are potentially associated with LngA protein variability, as discussed previously [12]. Recent studies have shown that the LngA protein, which assembles into CS21 pili, is highly immunogenic and may inhibit ETEC intestinal shedding [31]. The differences in the adherence percentages of the clinical ETEC strains and their position in the phylogenetic tree could also be attributed to the expression of other adhesins that promote adherence to HT-29 intestinal cells, such as type 1 fimbriae. In other bacteria, replacement of the *bfpA* gene (encoding the BfpA structural subunit of the BFP pilus of EPEC) by the *tcpA* gene (encoding the TcpA structural subunit of the TCP pilus) resulted in the production of an immature TcpA protein without complete processing by the prepilin peptidase BfpB; similarly, the bacteria were unable to assemble and produce the TCP pilus, which suggested that the two assembly mechanisms of the BFP and TCP pili are not compatible [32].

In *Neisseria meningitidis*, natural antigenic variation in the pilin PilE results in the bacterial strains showing differences in their aggregation ability [33]. ECBL carrying the *lng* operon could express the LngA protein and exhibited assembly of the CS21 pilus. The presence of variable regions in proteins with adhesion properties, such as the LngA protein, confers adaptive advantages in bacterial pathogenesis, including immune system evasion [29]. Other studies have shown that a variable region of 12 amino acids in the N-terminus of the BabA protein in *Helicobacter pylori* promotes different levels of adherence [34]. The M protein is the major virulence determinant of *Streptococcus pyogenes* and has a hypervariable region that gives it the ability to evade cellular phagocytosis [35,36].

Antigenic variations were observed at the predicted LngA surface accessible to B-cell epitopes, and the number of B-cell epitopes was sufficiently high to allow antigen recognition among different LngA variants [31]. To demonstrate whether variability in the C-terminus is related to the adherence of two ETEC (E9034A and FMU073332) strains previously described as strongly adherent, ETEC FMU07332 exhibited the highest adherence, and point mutations in which eight amino acids of the ETEC FMU07332 *lngA* gene were replaced by the E9034A *lngA* gene sequence were generated, cloned and transformed into ECBLΔ*lngA*. The substitutions of the amino acid residues GN by TT, G by S, TD by NN, and T by Q induced an important and significant reduction in adherence percentages compared with those in the E9034A and FMU73332 strains.

Our data indicate that the arrangement of amino acid residues such as T, K, G, and N could directly modify the interaction with possible ligands of their host, i.e., HT29 intestinal cells. In *Borrelia hermsii*, a high rate of amino acid variation in the VMP membrane protein is associated with evasion of the host immune system [37]. The accumulation of point mutations can be an adaptive functional tool used by different clinical ETEC strains to alter adhesion properties, as described with the BFP pilus of EPEC BFP and type 1 fimbriae from *Salmonella enterica* serovar Typhimurium [38].

The presence of amino acid residues with high variability at the C-terminus could be an essential strategy during the structural organization of the LngA protein that efficiently contributes to the correct assembly of the CS21 pilus; however, additional studies are needed.

The variability in the LngA proteins of clinical ETEC strains confers different attributes to the bacteria to promote colonization of intestinal cells and could confer different features to the bacteria to favor interaction with specific receptors located on the surface of HT-29 intestinal cells. These data indicate that CS21 has the potential to be considered a therapeutic target in the production of a potential vaccine. In conclusion, the LngA protein of clinical ETEC strains gives ECBL the ability to adhere to HT-29 intestinal cells, and the variability and point mutations in this protein modified their adhesion ability. These findings confirm the role of the CS21 pilus in the bacterial adherence process.

## Figures and Tables

**Figure 1 pathogens-12-00337-f001:**
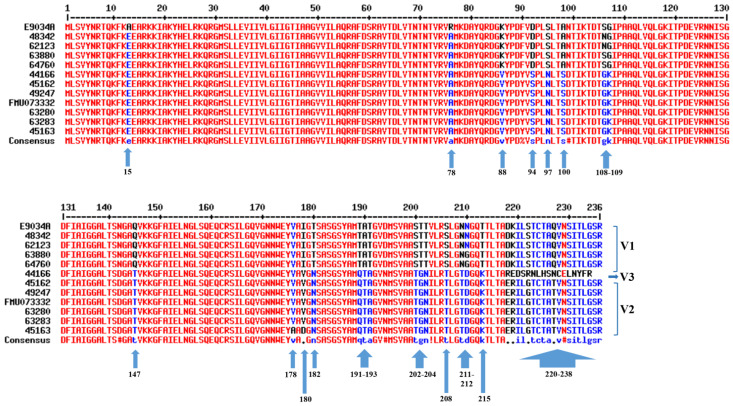
Multiple alignment of the amino acid sequences of the LngA variants of clinical ETEC strains. The consensus sequences are shown in different colors. High consensus is shown in red, low consensus is indicated in blue, and neutral consensus is indicated in black. Alignment of the LngA proteins was performed by multiple sequence alignment with hierarchical clustering (MultAlin program). The number on the top corresponds to the positions of the LngA amino acid residues. The arrows show the amino acid positions with changes, and the variants (V1–V3) were classified according to the changes in amino acid residues.

**Figure 2 pathogens-12-00337-f002:**
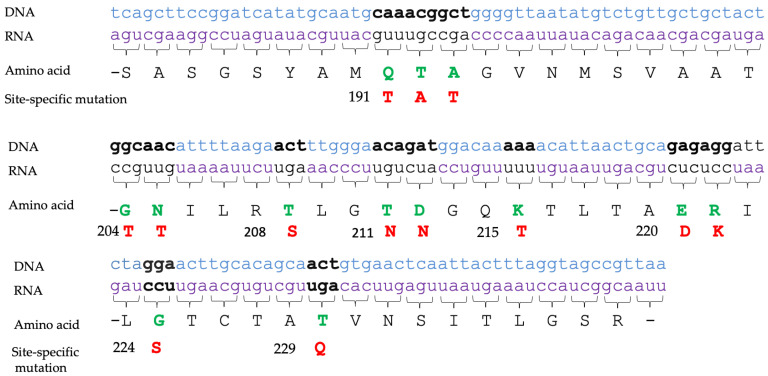
C-terminal site-specific mutations in the LngA sequence from FMU073332. The amino acid sequence for LngA of FMU073332 is shown in green and black. The site-specific mutation in the amino acid sequence of FMU073332 for E9034A is shown in green and red.

**Figure 3 pathogens-12-00337-f003:**
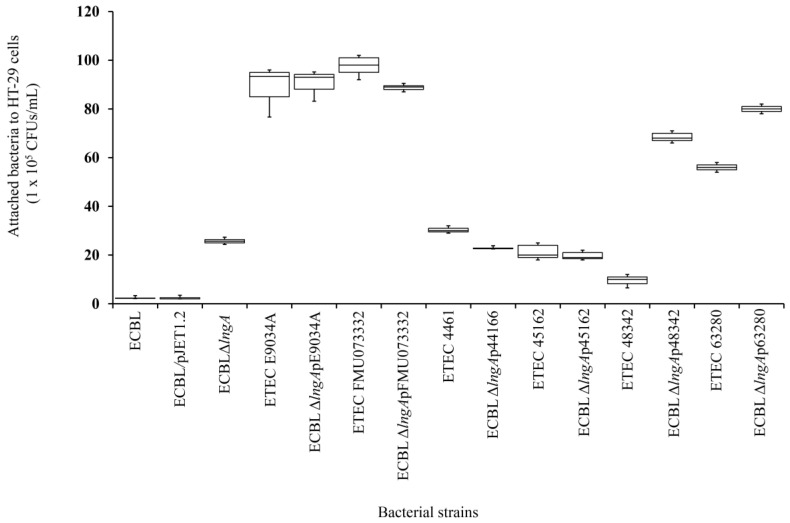
Quantitative analysis showed that recombinant ECBL strains with *lngA* variants could adhere to HT-29 cells. The quantitative analysis showed comparable profiles of clinical ETEC strains and recombinant ECBL strains with *lngA* variants, all of which restored the ability to adhere to HT-29 cells to levels similar to those achieved with ETEC. The adherent bacteria are expressed as CFUs after plating serial dilutions. The data are representative of at least three experiments performed in triplicate on different days. *p* ≤ 0.05 indicates statistical significance.

**Figure 4 pathogens-12-00337-f004:**
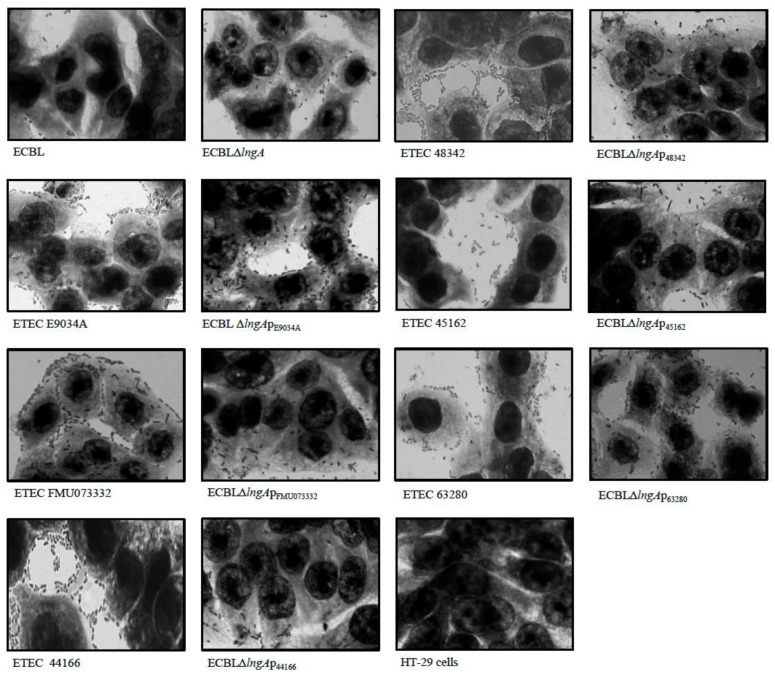
Qualitative analysis of ETEC E9034A and recombinant ECBL strains with *lngA* variants. Micrographs obtained by light microscopy showing bacteria adhering to monolayers of HT-29 cells with different levels, ETEC E9034A, FMU073332 and recombinant ECBL strains with *lngA* variants. HT-29 cells without infection were used as controls. Light micrography images were obtained at 100× magnification.

**Figure 5 pathogens-12-00337-f005:**
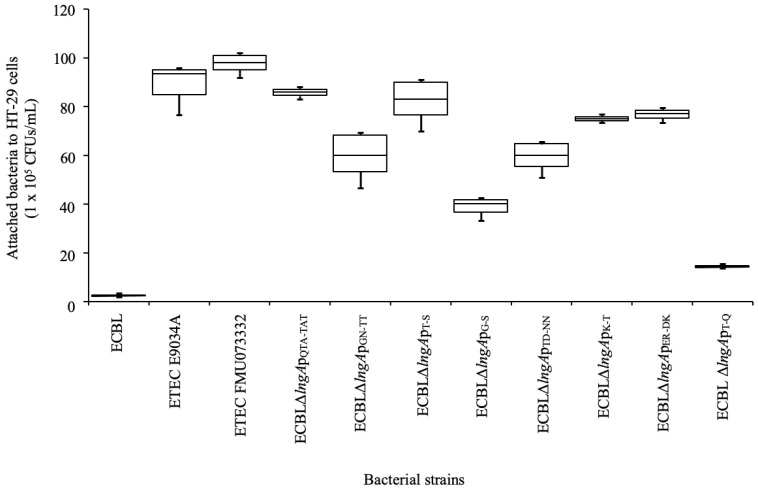
Quantitative analysis of the adherence of recombinant ECBL strains with site-specific mutations in the *lngA* gene to HT-29 intestinal cells. The adherence of recombinant ECBL strains with site-specific mutations to HT-29 intestinal cells was quantified in units of CFUs/mL after plating serial dilutions. The data represent at least three experiments performed in triplicate on different days. *p* ≤ 0.05 indicates a significant difference compared with the ETEC strains E9034A and FMU073332.

**Figure 6 pathogens-12-00337-f006:**
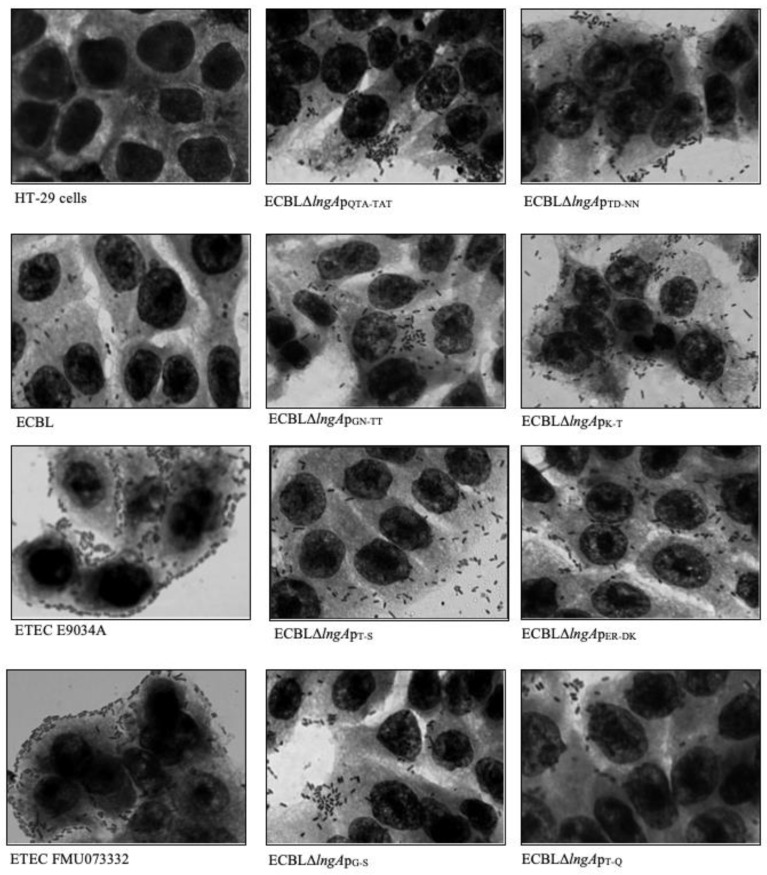
Qualitative analysis of recombinant ECBL strains with site-specific mutations. Micrographs obtained by light microscopy showing ETEC E9034A, FMU073332, and recombinant ECBL strains with site-specific mutations adhering to monolayers of HT-29 cells. The images were obtained at a magnification of 100× under a light microscope.

**Table 1 pathogens-12-00337-t001:** Information on previously characterized clinical ETEC strains.

ETEC Isolate	Origin	Serotypes	CS21	LT	ST	CS1	CS3	CFA/I	CS8	Adherence to HT-29 Cells (CFUs/mL)	
E9034A	Caribbean	O8:H9	+	+	+	-	+	+	+	4–8 * × 10^6^	Strong
44166	Mexico	O78:H12	+	-	+	-	-	+	-	3 × 10^6^	Moderate
45162	Mexico	O78:H12	+	+	+	-	-	-	-	2 × 10^6^	Weak
49247	Mexico	O6:H16	+	+	+	-	+	-	-	42 × 10^6^	Strong
62123	Mexico	O6:H16	+	+	-	-	-	-	-	40 × 10^6^	Strong
63280	Mexico	O78:H12	+	-	+	-	-	+	-	5.6 × 10^6^	Moderate
63880	Mexico	O6:H16	+	+	-	-	-	-	-	3.5 × 10^6^	Moderate
64760	Mexico	O8:H^−^	+	-	+	-	-	-	-	4.6 × 10^6^	Moderate
FMU073332	Mexico	O6:H16	+	+	+	-	+	-	-	8.5 × 10^6^	Strong
45163	Mexico	O78:H12	+	-	-	-	-	-	-	9.5 × 10^6^	Strong
48342	Mexico	O6:H16	+	+	+	+	+	-	-	0.95 × 10^6^	Weak
63283	Mexico	O78:H12	+	-	-	-	-	-	-	32 × 10^6^	Strong

Cruz-Cordova et al. [12]; * Saldana-Ahuactzi et al. [13]. Colonization factors (CS21, CS1, CS3, CFA/I and CS8) and toxins (LT and ST).

**Table 2 pathogens-12-00337-t002:** Description of strains used in this study.

Strain	Operon *lng*	Gen *lngA*	Kanamycin Resistance	Description	Reference
EBCL	-	-	-	*E. coli* BL21, suitable for transformation	Lab collection
EBCL/pJET 1.2	-	-	-	*E. coli* BL21 with pJET 1.2	This study
E9034A	+	+	-	Wild-type ETEC (O8:H9, CS21^+^, CS3^+^, STp^+^ and LT^+^)	Levine et al. [21]
FMU73332	+	+	-	Wild-type ETEC (O6:H16, CS21^+^, CS3^+^, STp^+^ and LT^+^)	Cruz-Cordova et al. [12]
E9034AΔ*lngA*::*km*	+	-	+	E9034A with a nonpolar insertional mutation in *lngA*	Cruz-Cordova et al. [12]

**Table 3 pathogens-12-00337-t003:** List of primers used in this study.

Primer	Sequence (5′→3′)	Use	Reference
LngA-F	TGCGGATCCGTGATCTGAAGAAAAATAA	Cloning of *lngA* in pJET1.2/blunt	Saldana-Ahuactzi et al. [13]
LngA-R	TGTGAGAAGGTACTAGCCTATCATATT		
LngAsec-F	CAGATTGGTTGAATCAGTTGTCA	*lngA* amplification and sequencing	This study
LngAsec-R	TGTGAGAAGGTACTAGCCTATCATATT		
LngH-F	AGAGAATTCCCGGGAAAGTACAGGCTG	Amplification of the *lngH* gene	Saldana-Ahuactzi et al. [13]
LngH-R	GAGTCATAGATCGGTAATCCTGAAAGCTTCAT		
lngD-F	GTCCCATGGGGATCCGTTTTCTTCAGAACAATAT	Amplification of the *lngD* gene	Saldana-Ahuactzi et al. [13]
lngD-R	CCATAAGAGCTCCAGCGCAATTTTTTCATC		
357	CTCCTACGGGAGGCAGCAG	16S amplification	Lane [22]
519	GWATTACCGCGGCKGCTG	16S amplification	

F: Forward; R: reverse.

**Table 4 pathogens-12-00337-t004:** Recombinant strains obtained after transformation of ECBLΔ*lngA* with the pJET1.2 vector carrying different ETEC variants of the *lngA* gene.

Recombinant Strain	Variant of *lngA* in ETEC Strains	Recombinant and Variant	Denomination	Description	Reference
ECBLΔ*lngA* *E. coli* BL21 carrying the plasmid with the *lng* operon of the CS21 pilus without the *lngA* gene (ECBLpE9034AΔ*lngA*)	E9034A	ECBLpE9034AΔ*lngA*/pJET_E9034A_	ECBLΔ*lngA*p_E9034A_	ECBL complemented with the *lngA* gene variant from the E9034A strain.	This study
	48342	ECBLpE9034AΔ*lngA*/pJET_48342_	ECBLΔ*lngA*p_48342_	ECBL complemented with the *lngA* gene variant from the 48342 strain.	This study
	44166	ECBLpE9034AΔ*lngA*/pJET_44166_	ECBLΔ*lngA*p_44166_	ECBL complemented with the *lngA* gene variant from the 44166 strain.	This study
	45162	ECBLpE9034AΔ*lngA*/pJET_45162_	ECBLΔ*lngA*p_45162_	ECBL complemented with the *lngA* gene variant from the 45162 strain.	This study
	FMU73332	ECBLpE9034AΔ*lngA*/pJET_FMU073332_	ECBLΔ*lngA*p_FMU073332_	ECBL complemented with the *lngA* gene variant from the FMU73332 strain.	This study
	63280	ECBLpE9034AΔ*lngA*/pJET_63280_	ECBLΔ*lngA*p_63280_	ECBL complemented with the *lngA* gene variant from the 63280 strain.	This study

**Table 5 pathogens-12-00337-t005:** Point substitution of amino acid residues in the LngA protein of ETEC. FMU73332 based on amino acid residues in the LngA protein of ETEC E9034A.

Recombinant Strain	Complementation Vector	Specific Mutation			Strains with Site-Specific Mutation
ECBLΔ*lngA*	pJET2.0 *lngA*_FMU073332_	Strain FMU73332	Amino Acid Substitution in ETEC E9034A	Localization	
ECBLΔ*lngA* *E. coli* BL21 carrying the plasmid with the *lng* operon of CS21 without the *lngA* gene (ECBLpE9034AΔ*lngA*)	Plasmid carrying the *lngA* gene of the FMU073332 strain	QTA	TAT	191–193	ECBLΔ*lngA*p_QTA-TAT_
		GN	TT	204, 205	ECBLΔ*lngA*p_GN-TT_
		T	S	208	ECBLΔ*lngA*_pT-S_
		G	S	224	ECBLΔ*lngA*p_G-S_
		TD	NN	211, 212	ECBLΔ*lngA*p_TD-NN_
		K	T	215	ECBLΔ*lngA*p_K-T_
		ER	DK	220, 221	ECBLΔ*lngA*p_ER-DK_
		T	Q	229	ECBLΔ*lngA*p_T-Q_

Glutamine (Q), threonine (T), alanine (A), glycine (G), asparagine (N), serine (S), aspartate (D), lysine (K), glutamic acid (E), and arginine (R).

## Data Availability

The datasets presented in this manuscript can be found in online repositories. The names of the repositories/repositories and accession number(s) can be found at the following link: https://github.com/ariadnnacruz/CS21-LngA.git (accessed on 5 December 2022).

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
