# Peer review of "Recombinant *Escherichia coli* BL21 with LngA Variants from ETEC E9034A Promotes Adherence to HT-29 Cells"

_pathogens, 2023, doi:10.3390/pathogens12020337_

Round 1
Reviewer 1 Report
There is a statement repeated twice between lines 465 and 469.
Author Response
Comments and Suggestions for Authors
There is a statement repeated twice between lines 465 and 469.
R/ One of the repeated statements has been removed from the manuscript.
Reviewer 2 Report
This manuscript confirm the role of the CS21 pilus in the bacterial adherence proces. The Authors investigated the effect of modyfied E. coli BL21 (ECBL) to adherence to HT-29 intestinal cells. The modifications encompassed in the first step the introducing of megaplasmid with lng operon of CS21 pilus without lngA gene and in the second step introducing the pJET vector with different variants of lngA gene of ETEC clinical strains as well as specific point substitutions in C-terminal region of lngA protein.
Major comments
The experiments were multilevel, well planned and consistently conducted. The results were properly presented. This is a very good job. But there are some fragments that need the corrections.
I also wonder why did the Authors choose the lngA gene to site-directed mutagenesis not lngB gene? Usually the tip of the pilus are mainly responible for adherence to intestinal cells.
Specific comments
Abstract
The abstract is a little chaotic and some fragments are difficult to understand to the readers.
Line 20
„This study aimed to determine whether E. coli BL21 (ECBL) carrying the variants of ETEC clinical strains……”
This sentence is mental shortcut and is difficult to understand. How ECBL can carry the ETEC strain? The Authors should explain what kind of ETEC variants?
In the abstract should appear the information what is the connection between CS21 pilus and lng operon and what kind of strain is E9034A?
Introduction
Line 37
„….cases of diarrhea and 120,800 deaths in 2010” – there are old data from 13 years ago. The Authors should provide more recent data.
Materials and Methods
Line 160
„In addition, electrocompetent cells from the ECBLΔlngA strain……” – My suggestions:
In addition, electrocompetent cells derived from the ECBLΔlngA strain…. Or
In addition, electrocompetent cells of the ECBLΔlngA strain…. Would be better
Line 195
Table 4 description is very complicated and unclear for the reader.
„Recombinant strains obtained from the lngA variant were cloned into the pJET1.2 vector and transformed into ECBLΔlngA.”
My suggestions:
Recombinant strains obtained after transformation of ECBLΔlngA with pJET1.2 vector carrying different ETEC variants of lngA gene
Table 4 and Table 5
First column:
My suggestion:
ECBLΔlngA E. coli carrying the megaplasmid with lng operon of the CS21 pilus without lngA gene (EC-BLpE9034AΔlngA)
Line 179
The Authors dont have to repeat the information point by point about site-directed mutagenesis. My suggestions „……were performed according to terminal carboxyl variability as presented in details in Table 5.
It would be better to change the text position in the third and fourth columns for example:
QTA -> TAT 191-193
GN -> TT 204, 205 and so on.
Please check the position 215-219, there are some discrepancies between the text (Line 182) and table 5.
Results
Line 340
Should be:
ECBLpE9034AΔlngA was transformed with pJETlngAFMU073332 vector variants to generate eight recombinant ECBL strains with site-specific lngA mutations:……
Line 363
„In contrast, mannose inhibition assays were performed to demonstrate that adherence was mediated mainly by LngA and not by type 1 fimbriae.”
Should be:
In addition, mannose inhibition assays were performed to demonstrate that adherence was mediated mainly by LngA and not by type 1 fimbriae.
Line 365
„The results showed a decrease in adherence with 1% mannose, which was significant compared with the results obtained from assays without mannose (p=0.00048; Figure S4).
Should be:
The results showed a significant decrease in adherence with 1% mannose (p=0.00048; Figure S4), compared with the results obtained from assays without mannose
Figure 3
The Authors should verify the data in table 1 with Figure 3.
In Figure 3 two strains ETEC E9034A and ETEC FMU073332 showed similar adherence to HT-29 Intestinal Cells. It seems 8x10^6 CFUs/mL for ETEC E9034A and 9.8x10^6 CFUs/mL for ETEC FMU073332 but not 98x10^6 like in table 1? In Figure 3 strain ETEC 63280 showed adherence about 5.5x10^6 CFUs/mL, but not 32x10r^6 as in table 1? Please verify all these datails.
Line 376
Why the Authors choose the substitution of the amino acid residues in the C-terminus of the LngA protein of the FMU073332 strain to amino acid residues in the LngA protein from the E9034A strain in experiment? These two strains ETEC E9034A and ETEC FMU073332 showed similar adherence to HT-29 Intestinal Cells, as we can see in Figure 3 and Figure 5. On the other hand, two strains ETEC E9034A with strong adherence and ETEC 48342 with weak adherence have the same amino acids sequence in C- terminal part of the LngA protein, but there is the difference in position 108-109, maybe this region is responsible for differences in adherence?
Lines 398-416
There is the repeat of subchapter 3.5. Recombinant ECBL Strains with Site-Specific Mutations Showed Modified Adherence to 372 HT-29 Intestinal Cells
Figure S5
The Authors should correct the lines captions and complete the description.
In Figure S5A upper part: there is the shift in lines captions, there is the product on the line 10, similarly in bottom part we can see the product on line 10, but in the description we have the negative control.
In Figure S5B in description there is: ECBLDlngA bacteria colonies to confirm lng operon insertion, but what type of colonies? Probably complemented with the lngA gene variant on pJET vector, because on the operon there is not lngA gene.
Discussion
Lines 468-470
The sentence „Because each pilus filament……” was repeated.
Line 475
„two regions were well defined (78 to 108 to 109 and 114),” – probably should be:
two regions were well defined (78 to 108 and 109 to 114)
Line 476
„According to 3D structure models of CofA of the CS8 pilus, the region from 189 to 236 of LngA has an exposed surface……..”- should be:
According to 3D structure models of CofA of the CS8 pilus, the corresponding region from 189 to 236 of LngA has an exposed surface……..”
Line 509
Typing error and should be in italics: In Neisseria meningitidis
Author Response
Comments and Suggestions for Authors
This manuscript confirms the role of the CS21 pilus in the bacterial adherence process. The Authors investigated the effect of modified E. coli BL21 (ECBL) on adherence to HT-29 intestinal cells. The modifications encompassed in the first step the introduction of megaplasmid with lng operon of CS21 pilus without lngA gene and in the second step introducing the pJET vector with different variants of lngA gene of ETEC clinical strains as well as specific point substitutions in C-terminal region of LngA protein.
Major comments
The experiments were multilevel, well planned and consistently conducted. The results were properly presented. This is an excellent job. Nevertheless, some fragments need corrections.
R/ Thank you very much for the comments.
Also, why did the authors choose the lngA gene to site-directed mutagenesis, not the lngB gene? Usually, the pilus tip is mainly responsible for adherence to intestinal cells.
R/ The reviewer's comment is correct, where the question arises as to why site-directed mutagenesis was not performed with the LngB protein. However, it is important to mention that we have generated several studies describing the participation of the LngA protein in ETEC adherence to intestinal cells. In addition, according to these preliminary data, these studies could contribute to the pathogenesis of ETEC by producing the CS21 pilus. We are working with LngB protein in other study; however, most studies are required to specifically determine the LngB role in adherence to intestinal cells.
Specific comments
Abstract
The abstract is a little chaotic, and some fragments are difficult for readers to understand.
R/ The abstract has been restructured as suggested by the reviewer.
Line 20
„This study aimed to determine whether E. coli BL21 (ECBL) carrying the variants of ETEC clinical strains……”
This sentence is a mental shortcut and is difficult to understand. How ECBL can carry the ETEC strain? The Authors should explain what kind of ETEC variants?
R/ The statement has been modified and completed for better understanding. The abstract has been restructured as suggested by the reviewer.
The abstract should appear with information on the connection between CS21 pilus and lng operon and what kind of strain it is E9034A?
R/ The statement has been modified for better understanding.
Introduction
Line 37
„….cases of diarrhea and 120,800 deaths in 2010” is old data from 13 years ago. The Authors should provide more recent data.
R/ The sentence was updated as suggested by the reviewer.
Materials and Methods
Line 160
„In addition, electrocompetent cells from the ECBLΔlngA strain……” – My suggestions:
In addition, electrocompetent cells derived from the ECBLΔlngA strain…. or electrocompetent cells of the ECBLΔlngAstrain…. Would be better
R/ The sentence was modified as suggested the reviewer 2.
Line 195
Table 4 description is very complicated and unclear for the reader.„
Recombinant strains obtained from the lngA variant were cloned into the pJET1.2 vector and transformed into ECBLΔlngA.”
My suggestions:
Recombinant strains obtained after transformation of ECBLΔlngA with pJET1.2 vector carrying different ETEC variants of lngA gene
R/ The sentence in Table 4 has been modified as suggested by the reviewer.
Table 4 and Table 5
First column:
My suggestion:
ECBLΔlngA E. coli carrying the megaplasmid with lng operon of the CS21 pilus without lngA gene (EC-BLpE9034AΔlngA).
R/ The sentences of Tables 4 and 5 have been modified as suggested by the reviewer.
Line 179
The Authors do not have to repeat the information point by point about site-directed mutagenesis. My suggestions „……were performed according to terminal carboxyl variability, as presented in Table 5.
R/ The changes were performed as suggested by the reviewer.
It would be better to change the text position in the third and fourth columns, for example:
QTA -> TAT 191-193
GN -> TT 204, 205 and so on.
Please check positions 215-219, there are some discrepancies between the text (Line 182) and table 5.
R/ All observations performed for the reviewer were approached.
Results
Line 340
Should be:
ECBLpE9034AΔlngA was transformed with pJETlngAFMU073332 vector variants to generate eight recombinant ECBL strains with site-specific lngA mutations:……
R/ The sentence was modified as suggested by the reviewer.
Line 363
„In contrast, mannose inhibition assays were performed to demonstrate that adherence was mediated mainly by LngA and not by type 1 fimbriae.”
Should be:
In addition, mannose inhibition assays were performed to demonstrate that adherence was mediated mainly by LngA and not by type 1 fimbriae.
R/ The sentence was modified as suggested by the reviewer.
Line 365
„The results showed a decrease in adherence with 1% mannose, which was significant compared with the results obtained from assays without mannose (p=0.00048; Figure S4).
Should be:
The results showed a significant decrease in adherence with 1% mannose (p=0.00048; Figure S4), compared with the results obtained from assays without mannose.
R/ The sentence was modified as suggested by the reviewer.
Figure 3
The Authors should verify the data in table 1 with Figure 3.
In Figure 3 two strains ETEC E9034A and ETEC FMU073332 showed similar adherence to HT-29 Intestinal Cells. It seems 8x10^6 CFUs/mL for ETEC E9034A and 9.8x10^6 CFUs/mL for ETEC FMU073332 but not 98x10^6 like in table 1? In Figure 3, strain ETEC 63280 showed adherence about 5.5x10^6 CFUs/mL but not 32x10r^6 as in table 1? Please verify all these details.
R/ Thank you very much for the observation. Yes, the data showed for ETEC FMU073332 in Table 1 has a mistake, it was revised and modified as 8.5 x106 CFUs/mL.
This manuscript reported to ETEC 63280 with an adherence of about 5.5x106 CFUs/mL, described in figure 3 and Table 1; while, in the article previously published by Cruz-Cordova et al., 2014, described an adherence value of 6.0 x106CFUs/mL.
Line 376
Why the Authors choose the substitution of the amino acid residues in the C-terminus of the LngA protein of the FMU073332 strain to amino acid residues in the LngA protein from the E9034A strain in the experiment? These two strains, ETEC E9034A and ETEC FMU073332, showed similar adherence to HT-29 Intestinal Cells, as we can see in Figure 3 and Figure 5. On the other hand, two strains ETEC E9034A with strong adherence and ETEC 48342 with weak adherence, have the same amino acids sequence in C- terminal part of the LngA protein, but there is a difference in position 108-109; maybe this region is responsible for differences in adherence?
R/ Thank you very much for this observation. The selection of these strains to perform the substitution of the amino acid residues in the C-terminus in the LngA protein of the ETEC FMU073332 strain was performed following the following points. First, in a previous study, ETEC E9034A and FMU073332 strains were compared according to their adhesion ability to intestinal cells and which FMU073332 strain was more adherent than E9034A strain. Second, the complete genome data are available for ETEC FMU073332 (NZ_CP017844.1), which was previously determined by our work group. In addition, only the lng plasmid of ETEC E9034A is also available at NCBI.
According to multiple alignments of the amino acid sequence of the LngA variants of clinical strains of ETEC, the amino acid residues at positions 178 to 238 of the C-terminus contain the main variable regions. From this data, this region could be involved in determining the adherence to intestinal cells. In addition, the substitution of amino acid residues in the LngA protein of the FMU073332 strain for the ETEC E9034A in this region could be improved the adherence in ETEC E9034A; in contrast, the result showed a reduction in adherence when two amino acids were modified.
Indeed, the difference at positions 108-109 of the LngA protein could be involved in the adherence differences, although most studies (example, computational biology) are required to demonstrate this hypothesis.
Lines 398-416
There is a repeat of subchapter 3.5. Recombinant ECBL Strains with Site-Specific Mutations Showed Modified Adherence to 372 HT-29 Intestinal Cells
R/ A repeat of subchapter 3.5 has been removed from the manuscript.
Figure S5
The Authors should correct the line captions and complete the description.
In Figure S5A upper part: there is a shift in line captions, and there is the product on line 10; similarly, in the bottom part, we can see the product on line 10, but in the description, we have the negative control.
In Figure S5B in the description, there is: ECBLDlngA bacteria colonies to confirm lng operon insertion, but what type of colonies? Probably complemented with the lngA gene variant on the pJET vector because, on the operon, there is no lngAgene.
R/ A Thank you very much. The sentence in the manuscript was modified.
Discussion
Lines 468-470
The sentence „ Because each pilus filament……” was repeated.
R/The first sentence repeated was removed as indicated by the reviewer.
Line 475
„two regions were well defined (78 to 108 to 109 and 114),” – probably should be: two regions were well defined (78 to 108 and 109 to 114)
R/ The comment of this reviewer has been carried out
Line 476
„According to 3D structure models of CofA of the CS8 pilus, the region from 189 to 236 of LngA has an exposed surface……..”- should be:
According to 3D structure, models of CofA of the CS8 pilus, the corresponding region from 189 to 236 of LngA has an exposed surface……..”.
R/ The sentence was modified as suggested by the reviewer
Line 509
Typing error and should be in italics: In Neisseria meningitidis
R/ The mistake was corrected
Reviewer 3 Report
- Line 232, bowtie2 is designed for illumina short reads not for long error-prone nanopore reads. The authors should use mappers that were optimized for long error-prone reads such as bwa, minimap2, blasr…
- Line 228 which config did the authors use when using guppy for basecalling? Different configs of guppy result in very significant differences in the error rates of the reads.
- Line 278 what does "La Jolla, CA, USA" mean? software or citation?
- Section 3.5 is duplicated. Did the author really take this manuscript seriously?
- Fig S2, some labels lanes don't match the lanes and need to be adjusted. Also, why is the size of last four 16S shorter than the others.
- The authors used nanopore sequencing to find the lng operon. Why not just use PCR and sanger sequencing, which is cheaper, faster, and more accurate than nanopore?
- The authors concluded that lngA impacts adhesion ability. Why is the ECBLΔlngA strain that has no lngA gene showing stronger adhesion than some strains with the lngA gene in Figure 3? In addition, ETEC 48324 and lngAp48342 have the same sequence of lngA gene, but the latter has much stronger adhesion than the former, which seems to indicate that pJET1.2 not the lngA gene is the factor that affects adhesion. How to explain this contradiction?
Author Response
Comments and Suggestions for Authors
1. Line 232, bowtie2 is designed for Illumina short reads, not for long error-prone nanopore reads. The authors should use mappers optimized for long error-prone reads such as bwa, minimap2, blasr…
R/ The idea of the sequencing was only to corroborate that the 14 kb plasmid and the pJET were within the strain. An important characteristic in that there is no upper limit on read length in Bowtie 2. This description was found in the user manual of bowtie2. The reference of this program is: Langmead, B. & Salzberg, S. L. Fast gapped-read alignment with Bowtie 2. Nat. Methods 9, 357–359 (2012).
2. Line 228 which config did the authors use when using guppy for base-calling? Different configs of guppy result in very significant differences in the error rates of the reads.
R/ The idea of the sequencing was only to corroborate that the 14 kb plasmid and the pJET 1.2 with lngA were into the strain, which is because default parameters were used.
3. Line 278, what does "La Jolla, CA, USA" mean? Software or citation?
R/ This mistake was corrected.
4. Section 3.5 is duplicated. Did the author really take this manuscript seriously?
R/ The section duplicated was removed; an apology for the oversight.
5. Fig S2, some label lanes do not match the lanes and need to be adjusted. Also, why is the size of the last four 16S shorter than the others?
R/ The mistake was corrected, and the original figure was inserted, showing the last four strains.
6. The authors used nanopore sequencing to find the lng operon. Why not just use PCR and sanger sequencing, which is cheaper, faster, and more accurate than nanopore?
R/ The plasmid is 14-kb for molecular weight, and we decided to sequence by nanopore because of the costs, it was cheaper than by sanger.
7. The authors concluded that lngA impacts adhesion ability. Why is the ECBLΔlngA strain with no lngA gene showing stronger adhesion than some strains with the lngA gene in Figure 3? In addition, ETEC 48324 and lngAp48342 have the same sequence of lngA gene, but the latter has much stronger adhesion than the former, indicating that pJET1.2, not the lngA gene, is the factor that affects adhesion. How to explain this contradiction?
R/ Effectively, the ECBLΔlngA strain with no lngA gene showed stronger adhesion than other strains because the LngB protein could participate in the adherence process to HT-29 cells. We considered that when LngA protein is interrupted, LngB protein is processed and exported to external cell membranes to participate in the adherence process. In the other strains, the LngA and LngB proteins are expressed and assembled as CS21 pilus.
The pJET1.2 has not been affecting adhesion because the effect is observed only in two strains, 48342 and 63280, neither all strains. Also, in figure 3 an additional control was included to demonstrate that it is not the effect of the plasmid pJET1.2. This result could be explained by the genetic background in ETEC 48342 related to regulation, when other proteins maybe display a negative regulation on CS21, compared to ECBLΔlngAp48342 genetic background.
Round 2
Reviewer 3 Report
R/ The idea of the sequencing was only to corroborate that the 14 kb plasmid and the pJET were within the strain. An important characteristic in that there is no upper limit on read length in Bowtie 2. This description was found in the user manual of bowtie2. The reference of this program is: Langmead, B. & Salzberg, S. L. Fast gapped-read alignment with Bowtie 2. Nat. Methods 9, 357–359 (2012).
R/ The idea of the sequencing was only to corroborate that the 14 kb plasmid and the pJET 1.2 with lngA were into the strain, which is because default parameters were used.
Q: It has nothing to do with the read length. Bowtie 2 was designed for the low-error short reads not for error-prone nanopore reads. That's why I asked for the parameters the authors used for base calling. The error rate of the reads by the guppy with default parameters is about 15%, which is too high to use Bowtie 2 to do mapping. In addition, the paper was released in 2012 while the first nanopore device released by oxford was 2014. Did the authors really read this paper? The authors don't seem to know much about bioinformatics, I doubt if they really did the analysis themselves. If the analysis was performed by others, let them to answer the questions.
R/ Effectively, the ECBLΔlngA strain with no lngA gene showed stronger adhesion than other strains because the LngB protein could participate in the adherence process to HT-29 cells. We considered that when LngA protein is interrupted, LngB protein is processed and exported to external cell membranes to participate in the adherence process. In the other strains, the LngA and LngB proteins are expressed and assembled as CS21 pilus.
Q: If the authors think LngB also participated in the adhesion, they should use appropriate control groups to eliminate the effect of the LngB.
The pJET1.2 has not been affecting adhesion because the effect is observed only in two strains, 48342 and 63280, neither all strains. Also, in figure 3 an additional control was included to demonstrate that it is not the effect of the plasmid pJET1.2. This result could be explained by the genetic background in ETEC 48342 related to regulation, when other proteins maybe display a negative regulation on CS21, compared to ECBLΔlngAp48342 genetic background.
Q: By the same logic, we can also think the pJET1.2 plasmid can affect adhesion because two strains were affected. The authors need appropriate control to eliminate the effect of pJET1.2.
Author Response
R/ The idea of the sequencing was only to corroborate that the 14 kb plasmid and the pJET 1.2 with lngA were into the strain, which is because default parameters were used.
Q: It has nothing to do with the read length. Bowtie 2 was designed for the low-error short reads not for error-prone nanopore reads. That's why I asked for the parameters the authors used for base calling. The error rate of the reads by the guppy with default parameters is about 15%, which is too high to use Bowtie 2 to do mapping. In addition, the paper was released in 2012 while the first nanopore device released by oxford was 2014. Did the authors really read this paper? The authors don't seem to know much about bioinformatics, I doubt if they really did the analysis themselves. If the analysis was performed by others, let them to answer the questions.
R/ We understood the reviewer's comment about the mapper program used and agree with his observation. However, we get similar results using the program minimap2 (minimap2 -t 24 -Q -x map-ont plasmids.fna minion.fasta > minionreads_vs_EF595770_minimap2.paf) to map the reads against the plasmid pE9034A (EF595770.1) and pJET 1.2 with lngA. Additionally, more than a thousand minion sequences map to the plasmids, covering more than 95% of them, showing that the plasmids are in the strain. Regarding the parameters with which we ran Guppy, and used the min_qscore flag to keep only reads with a quality score above 7 as recommended nanopore (Delahaye, Clara, and Jacques Nicolas. "Sequencing DNA with nanopores: Troubles and biases." PloS one 16.10 (2021): e0257521.).
R/ The idea of the sequencing was only to corroborate that the 14 kb plasmid and the pJET were within the strain. An important characteristic in that there is no upper limit on read length in Bowtie 2. This description was found in the user manual of bowtie2. The reference of this program is: Langmead, B. & Salzberg, S. L. Fast gapped-read alignment with Bowtie 2. Nat. Methods 9, 357–359 (2012).
R/ The idea of the sequencing was only to corroborate that the 14 kb plasmid and the pJET 1.2 with lngA were into the strain, which is because default parameters were used.
Q: It has nothing to do with the read length. Bowtie 2 was designed for the low-error short reads not for error-prone nanopore reads. That's why I asked for the parameters the authors used for base calling. The error rate of the reads by the guppy with default parameters is about 15%, which is too high to use Bowtie 2 to do mapping. In addition, the paper was released in 2012 while the first nanopore device released by oxford was 2014. Did the authors really read this paper? The authors don't seem to know much about bioinformatics, I doubt if they really did the analysis themselves. If the analysis was performed by others, let them to answer the questions.
R/ We understood the reviewer's comment about the mapper program used and we agree with his observation. However, we get similar results using the program minimap2 (minimap2 -t 24 -Q -x map-ont plasmids.fna minion.fasta > minionreads_vs_EF595770_minimap2.paf) to map the reads against the plasmid pE9034A (EF595770.1) and pJET 1.2 with lngA, we have more than a thousand minion sequences that map to the plasmids, covering more than 95% of it, which shows that the plasmids are in the strain. Regarding the parameters with which we ran Guppy, we actually use the min_qscore flag to keep only reads with a quality score above 7 as it is recommended for nanopore (Delahaye, Clara, and Jacques Nicolas. "Sequencing DNA with nanopores: Troubles and biases." PloS one 16.10 (2021): e0257521.)
R/ Effectively, the ECBLΔlngA strain with no lngA gene showed stronger adhesion than other strains because the LngB protein could participate in the adherence process to HT-29 cells. We considered that when LngA protein is interrupted, LngB protein is processed and exported to external cell membranes to participate in the adherence process. In the other strains, the LngA and LngB proteins are expressed and assembled as CS21 pilus.
Q: If the authors think LngB also participated in the adhesion, they should use appropriate control groups to eliminate the effect of the LngB.
R/ We appreciate the comments made by the reviewer and we are in the best disposition to clarify them. The adhesion inhibition assay was performed with ECBLDlngA and ECBLDlngA incubated 1h at 37ºC with anti-LngB 1:10 (antibodies was previously reported by Saldaña-Ahuactzi et al. 2016). The results showed a nearly halved reduction in adherence when ECBLDlngA was incubated with anti-LngB, suggesting the role of LngB in adherence in this strain.
The pJET1.2 has not been affecting adhesion because the effect is observed only in two strains, 48342 and 63280, neither all strains. Also, in figure 3 an additional control was included to demonstrate that it is not the effect of the plasmid pJET1.2. This result could be explained by the genetic background in ETEC 48342 related to regulation, when other proteins maybe display a negative regulation on CS21, compared to ECBLΔlngAp48342 genetic background.
Q: By the same logic, we can also think the pJET1.2 plasmid can affect adhesion because two strains were affected. The authors need appropriate control to eliminate the effect of pJET1.2.
R/ The pJET1.2 was transformed into ECBLDlngA to demonstrated pJET1.2 has not been affecting adherence, in this assay, the pJET1.2 reduced in 20% the adherence ability to HT-29.
R/ Effectively, the ECBLΔlngA strain with no lngA gene showed stronger adhesion than other strains because the LngB protein could participate in the adherence process to HT-29 cells. We considered that when LngA protein is interrupted, LngB protein is processed and exported to external cell membranes to participate in the adherence process. In the other strains, the LngA and LngB proteins are expressed and assembled as CS21 pilus.
Q: If the authors think LngB also participated in the adhesion, they should use appropriate control groups to eliminate the effect of the LngB.
R/ We appreciate the reviewer's comments and are in the best position to clarify them. The adhesion inhibition assay was performed with ECBLlngA, and ECBLDlngA incubated for 1 h at 37ºC with anti-LngB 1:10 (antibodies were previously reported by Saldaña-Ahuactzi et al. 2016). The results showed a nearly halved reduction in adherence when ECBLDlngA was incubated with anti-LngB, suggesting the role of LngB in adherence in this strain.
Q: By the same logic, we can also think the pJET1.2 plasmid can affect adhesion because two strains were affected. The authors need appropriate control to eliminate the effect of pJET1.2.
R/ The pJET1.2 was transformed into ECBLDlngA to demonstrate pJET1.2 has not been affecting adherence, in this assay, the pJET1.2 reduced in 20% the adherence ability to HT-29.
